# Effect of Pretreatments and Drying Methods on Physical and Microstructural Properties of Potato Flour

**DOI:** 10.3390/foods11040507

**Published:** 2022-02-10

**Authors:** Ariel Buzera, Evelyne Gikundi, Irene Orina, Daniel Sila

**Affiliations:** 1Faculty of Agriculture and Environmental Science, Université Evangélique en Afrique (UEA), Bukavu P.O. Box 3323, Sud-Kivu, Democratic Republic of the Congo; 2Department of Food Science and Technology, Jomo Kenyatta University of Agriculture and Technology (JKUAT), Nairobi P.O. Box 62000-00200, Kenya; nkirote26@gmail.com (E.G.); irene.orina@jkuat.ac.ke (I.O.); dsila@jkuat.ac.ke (D.S.)

**Keywords:** potato flour, pretreatments, drying methods, particle size distribution, flow characteristics, microstructural properties

## Abstract

This study evaluated the effects of pretreatments (blanching (60 and 95 °C) and boiling) and drying methods (freeze-drying and oven drying) on the quality characteristics of potato flour derived from three potato varieties, namely, Shangi, Unica, and Dutch Robjin. The percentage flour yield, color, particle size distribution, flow characteristics, microstructural and functional properties of the potato flour were determined. Unica recorded the least peeling loss, while the Dutch Robjin variety had the highest. Color parameters were significantly affected (*p* < 0.05) by the pretreatments and drying methods. Freeze drying produced lighter potato flour (L* = 92.86) compared to the other methods. Boiling and blanching at 95 °C followed by oven drying recorded a low angle of repose and compressibility index, indicating better flow characteristics. The smallest particle size (56.5 µm) was recorded for the freeze-drying treatment, while boiling followed by oven drying had the largest particle size (307.5 µm). Microstructural results indicate that boiling and blanching at 95 °C, followed by oven drying resulted in damaged starch granules, while freeze-drying and low-temperature blanching (60 °C) maintained the native starch granule. Particle size and the solubility index of potato flour showed strong positive correlation. This study revealed that the pretreatments and drying methods affected potato flour’s physical and microstructural parameters differently, resulting in changes in their functionality.

## 1. Introduction

The potato (*Solanum tuberosum* L.) is considered to be an important food crop after rice, wheat, and maize [1]. It constitutes an excellent energy and dietary fiber source whose nutritional characteristics are essential in human consumption [2]. The second most important food crop in Kenya after maize is the potato [3]. It is both a staple food and a great source of income for many households (MoA, 2007). The average production in Kenya is estimated at 7 to 10 tons per hectare [4], compared to a global average yield of 17 tons per hectare [5]. The per capita consumption in Kenya is estimated at 31.8 kg annually [6]. Postharvest losses in Kenya are estimated to be 12.8% at the farm, 24.4% at the open market, and 12% at the processing facilities [7]. Most of these losses are due to the lack of optimal storage facilities, reported for most developing countries [8].

Once harvested, potatoes are destined to be used for diverse applications. Less than half of harvested potatoes are consumed fresh [9]. The rest are processed into food products or food ingredients [10]. Fresh potatoes are boiled, baked, fried, or used in a number of recipes (mashed potatoes, potato soup, and potato salad). During processing, potatoes are transformed into products that are less bulky, less perishable, and less expensive to store and transport [11]. Additionally, the consumption of fresh potatoes is shifting to added-value processed potato products [12]. Some of the new potato-derived food products include bread [13] and pasta [14]. Other potato-based products, such as snacks [15], noodles [16], potato chips [17,18], and potato flour [14], are increasingly being produced. This has expanded the utilization window of potatoes in food and industrial applications.

Processing potatoes into flour produces not only a product that is nutritionally and functionally adequate, but also shelf-stable [19]. The protein content of potato flour has a well-balanced amino acid composition, which can palliate the deficiency of cereal protein [20]. In addition, the lysine content of potato flour is close to that of animal protein; thus, potato flour can be used to overcome deficiencies in protein and calories [21]. Besides the nutritional importance, potato flour can be incorporated into various food products [22,23] such as bread, as it aids in retaining bread freshness [24]. It also gives a remarkable, distinctive flavor and improves the toasting qualities of bread [25]. Furthermore, the low water content, long shelf life, and transportation capability make potato flour suitable for use in a wide range of situations [26].

Different processes are involved in potato flour preparation [26]. Drying is one of the most commonly used processes [27]. Processing potatoes into flour can affect the physicochemical properties of the flour depending on the choice and conditions of the processing methods [28]. Consequently, based on the target application, drying methods and preprocessing conditions are essential factors in determining the physical and chemical properties of potato flour. Several studies have described the effects of drying on the quality attributes of roots and tubers such as sweet potatoes [29], cassavas [30], yams [31], red cocoyams [22], carrots [32], beetroots [33]. However, there is limited information on the correlation among the potato flour’s physicochemical, microstructural, and functional properties derived from different drying methods. Therefore, the objective of this study was to determine the effect of different pretreatments (blanching, boiling) and drying techniques (oven drying and freeze-drying) on the physical, microstructural, and functional properties of potato flour.

## 2. Materials and Methods

### 2.1. Plant Material

Three popular potato varieties (*Solanum tuberosum* L.) grown in Kenya, namely, Shangi, Unica, and Dutch Robjin, were procured locally from Nyandarua county. These varieties are the most commonly consumed and used in diverse applications in industries [34]. They also have high tuber yields [35].

### 2.2. Percentage Peel Loss

One kilogram of cleaned potato tubers from each variety was weighed and peeled manually using a sharp stainless kitchen knife. The peeled potatoes were then weighed. The percentage peel loss was calculated as shown in Equation (1) [36].
(1)$$\mathrm{Amount}\;\mathrm{of}\;\mathrm{peel}\;(\%)=\frac{\mathrm{WBP}-\mathrm{WAP}}{\mathrm{WBP}}\;\times 100$$
where WBF, weight of potatoes before peeling, WAP, weight of potatoes after peeling.

### 2.3. Preparation of Potato Flour

#### 2.3.1. Oven Drying

One kilogram of each of the three potato varieties was washed and manually peeled with a sharp stainless-steel knife. The peeled potatoes were cut into thin slices and then kept in water containing 0.5% sodium metabisulfite (SMB) for 5 min to prevent enzymatic darkening [37]. The thin slices were placed in an oven drier (Memmert UF 110 model, Schwabach, Germany) at 50 °C for 48 h with a constant air-flow rate of 2 m/s. The dried slices were finely ground into flour using a blender (Vitamix 5200 Blender, professional-grade, Cleveland, OH, USA). The flour was passed through a 500 μm sieve, packed in transparent polyethylene zip-lock bags, and stored at room temperature until further analysis.

#### 2.3.2. Blanching and Oven Drying

Blanched potato flours were prepared according to the methods described by [32], with a slight modification. First, clean potato tubers were washed with tap water, manually peeled, and thinly sliced into 2 mm thickness. Next, potatoes slices were blanched in water at 60 °C (low temperature) for 30 min and another portion at 95 °C (high temperature) for 1 min, and immediately cooled in an ice water bath for 5 min. The blanched samples were then oven-dried as described in Section 2.3.1, finely ground, sieved through a 500 μm standard sieve, and packed in transparent polyethylene zip-lock bags, and stored at room temperature until further analysis.

#### 2.3.3. Boiling and Oven Drying

Boiled potato flour was prepared using a method as previously described [38]. Washed and peeled potatoes tubers were boiled in tap water at ~95 °C for 30 to 35 min and immediately cooled in an ice water bath for 5 min. Boiled potatoes were mashed using a conventional potato masher and placed into the oven drier (Memmert UF 110 model, Schwabach, Germany) at 50 °C for 48 h. The dried mashed potatoes were ground, passed through a 500 μm standard sieve, and stored in polyethylene zip-lock bags at ambient temperature before further analysis.

#### 2.3.4. Freeze-Drying

This was conducted as described by [39], with slight modifications, using a freeze-dryer (Lyovapor L-200 Pro, BUCHI, Flawil, Switzerland). One kilogram of cleaned potato tubers was manually peeled and thinly cut into 2 mm thickness slices. Slices were placed in zip-lock bags pierced with holes and frozen in a deep freezer at −21 °C for 24 h before placing them in a freeze dryer. The holes allowed for regulating temperature and pressure inside and outside the zip-lock bags during the drying process. Initial drying was carried out at −41 °C and 0.11 millibar, while final drying was carried out at −47 °C and 0.055 millibars. The freeze-drying process lasted 48 h in total. The freeze-dried potato slices were then collected, crushed, and milled in a blender before storage for further analysis.

### 2.4. Percentage Yield of Potato Flour

One kilogram of each potato variety was washed and manually peeled with a sharp stainless-steel knife. After peeling, potatoes underwent pretreatments to produce flour, as explained in Section 2.3. The amount of flour from 1 kg of peeled potato was weighed. The percentage yield of potato flour was calculated as shown in Equation (2) [14].

### 2.5. Physical Properties Determination

#### 2.5.1. Potato Flour Color Determination


(2)
$$\mathrm{Potato}\;\mathrm{flour}\;\mathrm{Yield}\;(\%)=\frac{\mathrm{Weight}\;\mathrm{of}\;\mathrm{potato}\;\mathrm{flour}\;\mathrm{after}\;\mathrm{Drying}}{\mathrm{Weight}\;\mathrm{of}\;\mathrm{whole}\;\mathrm{peeled}\;\mathrm{potato}}\times 100$$


Potato flour color was measured using a HunterLab’s ColorFLex EZ spectrophotometer. The instrument was first calibrated by using white and black tiles as the standards. Next, potato flour was tightly packed in an optically transparent glass cup and covered with another opaque cover, used as a light trap to prevent the external light from interfering with the sample cup. The clear glass containing the flour was then placed on the port. Light from the spectrophotometer was flashed on the sample, and the color intensity was recorded. The potato flour color was reported in terms of 3-dimensional color values on the following rating scale: Lightness L* ranges from (black (0) to light (100)), a* from red (60) to green (−60), and b* ranges from yellow (60) to blue (−60). Measurements were conducted in triplicate. a* and b* values were used to calcite values for hue angle (H*) and chroma (C*) (Equations (3) and (4)), two parameters that are used for describing the visual color appearance [40].
H* = tan^−1^(b/a)(3)
C* = (a^2^ + b^2^)^−1^(4)

#### 2.5.2. True Density

True density was calculated according to [41]. Toluene (C_7_H_8_) was used instead water because it is absorbed by flour to a lesser extent. Approximately 1 g of potato flour was filled in a measuring cylinder containing toluene. The rise in toluene level was measured twice. True density was calculated as indicated in Equation (5):(5)$$\mathrm{True}\;\mathrm{density}\;(g/\mathrm{mL})=\frac{\mathrm{weight}\;\mathrm{of}\;\mathrm{flour}\;\mathrm{sample}}{\mathrm{rise}\;\mathrm{in}\;\mathrm{toluene}\;\mathrm{level}}$$

#### 2.5.3. Bulk Density and Tapped Density

The bulk and tapped density of potato flour were determined according to the method described by [42]. Potato flour (10 g) was filled into a 100 mL measuring cylinder, and bulk volume (V) was recorded. The bottom of the cylinder was continuously tapped (100 taps) on the platform, and the volume was recorded as tapped volume (V_T_). Bulk and tapped density were calculated as shown in Equations (6) and (7):(6)$$\mathrm{Bulk}\;\mathrm{Density}\;(B_{D})=\frac{Mass}{V}$$
(7)$$\mathrm{Tapped}\;\mathrm{Density}\;(B_{T})=\;\frac{Mass}{VT}$$

#### 2.5.4. Compressibility Index

The compressibility index of potato flour was determined on the basis of bulk density and tapped density as described by [42] (Equation (8)).
(8)$$\mathrm{Compressibility}\;\mathrm{index}\;(\%)=\frac{\mathrm{Tapped}\;\mathrm{density}-\mathrm{Bulk}\;\mathrm{density}}{\mathrm{Tapped}\;\mathrm{density}}$$

#### 2.5.5. Angle of Repose

The angle of repose (θ) was determined by using the method described by [42]. A funnel was mounted on a laboratory stand at the height of 2 cm from the bench. Potato flour (10 g) was weighed and allowed to flow through the funnel to form a pile at the base. The tip plug was removed, and the flour was allowed to pass through the orifice. The height and diameter of the flour heap were measured. Angle of repose θ was calculated as shown in Equation (9):(9)$$\left(\theta )=\tan^{-1}\;(\frac{H}{R\;}\right)$$
where *H* is the height of the cone formed after the flow was complete, and *R* is the radius of the cone.

### 2.6. Particle Size Distribution of Potato Flour

Particle size distribution was determined as described by [43]. A laser diffraction particle size analyzer (SALD-2300; Shimadzu Corporation, Kyoto, Japan) equipped with a cyclone injection unit (SALD-2300 Cyclone Injection Type Dry Measurement Unit SALD-DS5) was used. The device is capable of measuring particle sizes in the range from 17 nm to 2500 μm. A small amount of flour was placed into a hopper and sucked across the laser beam by pressing an ejector. Particle size distribution was determined by the light intensity distribution pattern of scattered light generated by a sample irradiated with a laser. Results were analyzed using Wing SALD II software (version 3.1.0, Shimadzu, Kyoto, Japan).

### 2.7. Microstructural Properties

#### 2.7.1. Light Microscopy

The potato flour was observed using a polarized optical microscope (B-1000 series, OPTIKA Microscope 24010, Ponteranica (BG), Italy) equipped with a camera (Optikam HDMI Pro Camera). A thin layer of potato flour was placed on a microscope slide and dispersed using a drop of distilled water. All images were observed and photographed at 20× magnification under an X-LED white illumination.

#### 2.7.2. Scanning Electron Microscopy

Flour micrographs were taken using a scanning electron microscope (model JCM-7000 NeoScope Benchtop SEM (JEOL Ltd., Tokyo Japan)). Potato flour was suspended on an aluminum stub using double-sided adhesive tape. Accelerating potential of 15 kV and magnification of 300× were used during micrography.

### 2.8. Solubility Index of Potato Flour

The solubility index of potato flour was determined following the method of [14], with slight modification. Potato flour (1 g, dry basis) in 10 mL of distilled water was heated to the desired temperature (50, 60, 70, 80, and 90 °C) for 30 min in a shaking water bath, and then centrifuged at 1600 rpm for 15 min. The supernatant was carefully collected in a preweighed dish and evaporated at 100 °C until constant weight. This was carried out in triplicate. Solubility was calculated as shown in Equation (10):(10)$$\mathrm{Solubility}\;\mathrm{index}\;(g/100\;g)=\frac{g\;\mathrm{Water}\;\mathrm{soluble}\;\mathrm{matter}\;\times 100}{g\;\mathrm{of}\;\mathrm{dry}\;\mathrm{sample}}$$

### 2.9. Statistical Analysis

All analyses were carried out in triplicate and expressed as means of standard errors. Data were subjected to statistical analysis using R-Console software. Results were analyzed using a one-way analysis of variance (ANOVA). In addition, the least-significance test (LSD) set at the 5% probability level was applied.

## 3. Results

### 3.1. Percentage Peel Loss (%)

The Dutch Robjin variety recorded the highest percentage peel loss (25%), while Unica showed the least (8%) (Figure 1). This difference could be attributed to the morphological differences between the varieties, especially the number and depth of eyes [44]. The Dutch Robjin variety had more eyes as compared to the other varieties. The more eyes the tuber has, and the deeper they are, the greater the peeling loss [45].

### 3.2. Percentage Yield of Potato Flour

Potato flour yields from the different potato varieties as affected by pretreatments and drying methods are shown in Table 1.

There was a significant difference in the yield of potato flour obtained from the different processing methods. Boiling followed by oven drying resulted in the lowest flour yield in all the three varieties. This might have been associated with the leaching of starch during boiling. On the other hand, the Dutch Robjin resulted in the highest flour yield in all the processing conditions. This might be attributed to its high specific gravity and dry matter compared to the other varieties [34,36,45]. There was no significant difference in yields of the flour obtained from oven drying and freeze-drying.

### 3.3. Color of Potato Flour

The color parameters of the potato flour varied greatly depending on the different pretreatments and drying methods (Table 2 and Figure 2). Only results for the Shangi variety are shown because similar trends were observed for Unica and Dutch Robjin.

Freeze-dried flour recorded the highest value of L* (92.86) and the lowest a* (−0.65) value. A higher L* value indicates white flour and a lower browning index [46]. Flours obtained from boiling followed by oven drying had the lowest L* value, and highest a* and b* values. Murayama [47] reported that a higher b* value indicates higher yellow color due to prolonged boiling or steaming during the preparation of potato flours. These would affect the final color of food products [29]. The higher a* value in boiled flour could have been due to the more prolonged exposure of samples to heat [48]. The hue angle (H*) was negative for freeze-dried flour. A negative value of the hue angle (H*) indicates that white was dominant in the sample. This was also reported by Olatidoye [49]. Chroma value is a measure of the color purity in a material [50]. It defines the color intensity of materials, so the higher the value of chroma is, the more intense the color [51]. Freeze drying, oven drying, and blanching at low temperature followed by oven drying led to the lowest chroma values compared to the high-temperature treatments (boiling and blanching at 95 °C followed by oven drying). Similar results were obtained by do Nascimento [52] on potato flour. A low value of chroma and a high value of lightness of flours are desired by consumers [53]. Therefore, freeze-dried flours would be more acceptable by consumers compared to flours from other processing methods.

### 3.4. Bulk, Tapped, and True Density

There was a significant difference (*p* < 0.05) in the bulk density of the potato flour prepared under different conditions (Table 3). Only results for the Shangi variety are shown. Bulk density varied from 0.49 g.mL^−1^ for freeze-dried flour to 0.91 g.ml^−1^ for boiling, followed by oven drying. Bulk and tapped densities give information on the particle packing and arrangement, and the material’s compaction profile [54]. They are influenced by particle size, particle size distribution, attractive interparticle forces, and particle shape [55].

The low bulk density of the freeze-dried flour might have mainly occurred because of the increased volume rather than the mass. The solubility of freeze-dried flour might significantly be affected by low bulk density [54].

Tapped density is determined to correct the fluctuation of the total volume of interparticle voids during transportation, drying, and packing processes. It is the maximal packing produced under the impact of an externally applied force. It indicates the volume of a mass of the sample after tapping the container to cause a closer packing of particles. Consequently, freeze-dried potato flour was more porous, since it had the lowest tapped density. This is because freeze-dried powder granules have small particles that can agglomerate to form a clumpy powder, thus producing more voids between them [56]. The consumption of food items with low bulk density is nutritionally important as it encourages the consumption of lighter food items [57]. These flours tend to have high viscosity, texture, and consistency that allow for easy consumption and digestibility (WHO, 2003). Boiling and blanching at high temperature flours showed the highest tapped density, thus indicating the least porosity among all samples. The high bulk density of flour suggests the suitability in food preparations with a suitable mixing ability, as reported by Chandra et al. [58]. In addition, these flours do not take up much space and decrease packaging costs. This study indicated that pretreatments and drying methods significantly (*p* < 0.05) influenced the true density of potato flour. The true density of the flours varied from 1.46 g/mL to 1.65 g/mL. Freeze-dried flour had the lowest true density compared to other treatments. This could be attributed to the ice sublimation during freeze-drying which creates more pores within the samples [59].

### 3.5. Compressibility Index (CI) and Angle of Repose (AoR)

The effects of different pretreatments and drying methods on compressibility index and angle of repose of potato flour are shown in Table 4. Only results for the Shangi variety are shown. The different flours showed different compressibility indices and angles of repose, which indicate different flow behavior.

The compressibility index ranged from 13.48% to 31.56% for boiling, followed by oven drying and freeze-drying, respectively. Boiling and blanching at high temperature (95°C) showed a minor compressibility index meaning that the flour has good flowability characteristics. Conversely, oven drying and freeze-drying had the highest compressibility indexes indicating poor flow behavior (cohesiveness). Flours with good flow behavior are suitable for easier mixing, transport, and other manufacturing manipulations [61].

The highest angle of repose was recorded for freeze-drying treatment while boiling and blanching at high temperature showed the lowest angle of repose. The angle of repose was significantly (*p* < 0.05) influenced by the pretreatments and drying methods, thus affecting the flow behavior. The angle of repose measures the flour resistance to the flow under gravity due to frictional forces resulting from the surface of the granules [62]. A high angle of repose implies a decrease in flowability characteristics [54]. The angle of repose is important for designing processing, storage, and conveying systems of particulate materials. A high angle of repose is associated with a fine and sticky material that is not free-flowing. In contrast, materials with a low angle of repose are highly flowable and can be transported using little energy [54].

### 3.6. Particle Size Distribution

The results for particle size and particle size distribution are presented in Figure 3 and Table 5. Only results for the Shangi variety are shown. The average diameter size of the potato flour ranged from 56.51 to 307.53 μm for freeze-dried potato flour and boiling followed by oven drying, respectively. This is in agreement with Daudt [63], who reported that cooked flour granules have larger particle sizes and a larger distribution of sizes, which might be attributed to the swelling of starch granules during heat treatment.

Unimodal particle size distribution was observed for freeze-drying, blanching at high temperature, and boiling, followed by oven-drying samples. Normal distribution (one peak) indicates good sample homogeneity and less interference [64]. On the other hand, bimodal distribution curves were observed for oven-dried flour and low-temperature blanching followed by oven drying. This indicates the presence of two clusters of samples, a cluster of small samples and a cluster of large samples. Figure 3b illustrates the cumulative frequency curves for the different types of flour, starting from the smallest (freeze-dried) to the largest (boiled). This is in agreement with the results in Table 5. Heating at high temperatures seemed to shift the size of particles from small to large. There are marked differences in the size of the particles per each percentile change in the cumulative frequency curve (Table 5) as indicated for D_10_ to D_90_. This might point to stark differences in the morphological and functional properties of the flours.

### 3.7. Light Microscopy (LM) and Scanning Electron Microscope (SEM)

LM and SEM images of the potato flours prepared under different pretreatments and drying methods are shown in Figure 4. Only results for the Shangi variety are shown. As seen in the light microscope images and SEM micrographs, the starch granules were oval-shaped for oven-dried flour, blanching at low temperature (60°), and freeze-dried flour. This is in line with previous research reported by Joyner and Meldrum [65], who mentioned that freeze-dried granules contained no large holes and had typical shapes of native potato starch. Boiling and blanching at high temperature resulted in larger and agglomerated starch granules. These findings agree with Nascimento [66], who reported that starch gelatinization and the exposure of heat to the internal starch surface cause a disruption of starch, which results in swelling and eventually irregularly shaped particles. These results confirm the observed differences in particle size and particle size distribution. Heating improves the hydration of cooked flours that might influence the pasting viscosity properties [23].

### 3.8. Solubility Index of Potato Flour

The solubility index of potato flour as influenced by different treatments and drying methods is presented in Figure 5.

The solubility of the flour increased with increasing temperature for all the samples. There was a strong positive correlation for solubility with increasing temperature (R^2^ > 0.98). The highest solubility index was observed for high-temperature processed samples (boiling and blanching at 95 °C) compared to freeze-dried and low-temperature blanching flour. These results agree with Kim and Kim [67], who reported an increase in solubility of potato flour with increasing temperature. The high solubility values reported in this study would be attributed to the high soluble content and leaching of amylose from starch [68]. Song et al. [69] reported that if flour or starch shows high solubility compared to others, it contains large amounts of soluble materials. High temperatures weaken the forces of attraction between flour particles, increasing the dissolution rate in water [70]. Strong positive correlation was also observed between particle size and the solubility index of potato flour (Figure 6).

With the increasing particle size of potato flour, solubility increased. This much varied, depending on the temperature at which the solubility of the particles was measured, resulting in parallel correlations lines with the highest solubility being seen for the larger particle for a given temperature. Larger particles result in higher solubility, as indicated by the solubility contours. Heat treatment disrupts granules’ structure and speeds up the leaching process of starch in water, increasing the solubility index [71]. Bala et al. [72] reported an increase in solubility with increasing particle size for grass pea flour.

## 4. Conclusions

In this study, the Unica variety resulted in the least peeling loss, while Dutch Robjin yielded the highest flour per kg of peeled potato. Freeze-drying resulted in the highest flour yield while boiling followed by oven drying resulted in the lowest. Freeze-drying and low-temperature blanching resulted in lighter-colored flour as opposed to boiling and high-temperature blanching (95 °C). Boiling and blanching at 95 °C treatments resulted in a low angle of repose and Carr’s index, meaning that they had good flow characteristics, suitable for easier mixing, transport, and further handling. Particle size, temperature, and the solubility index of potato flour were strongly correlated. Potato flour derived from different processing methods depicted various physical and morphological properties, indicating significant variations in the application window in the food and pharmaceutical industries.

## Figures and Tables

**Figure 1 foods-11-00507-f001:**
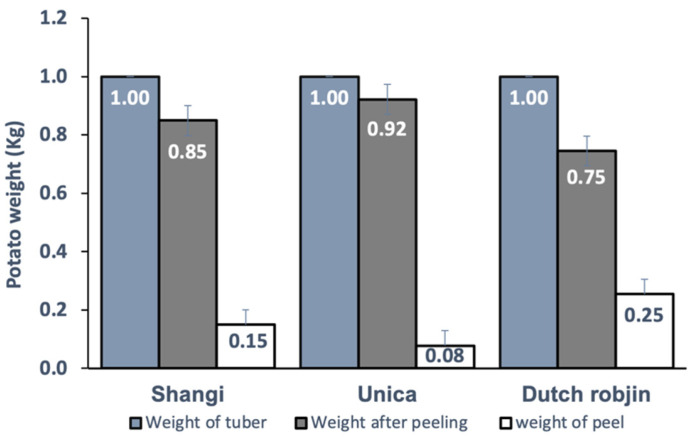
Percentage peeling loss of the three varieties (Shangi, Unica, and Dutch Robjin).

**Figure 2 foods-11-00507-f002:**
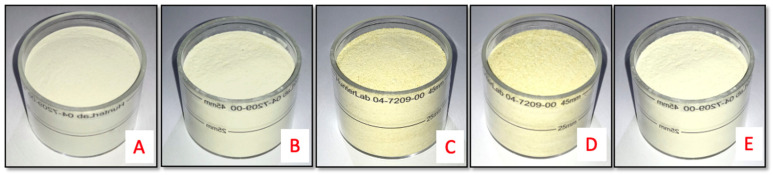
Potato flour prepared from different pretreatments and drying methods. (**A**) Oven drying; (**B**) blanching at low temperature (60 °C); (**C**) blanching at high temperature (95 °C); (**D**) boiling; (**E**) freeze drying.

**Figure 3 foods-11-00507-f003:**
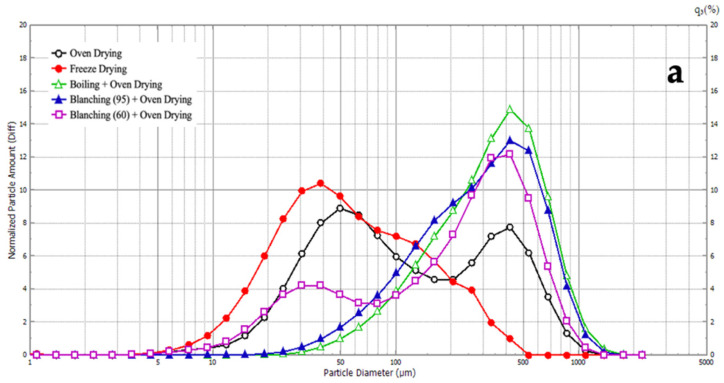
(**a**) Particle size distribution curves and (**b**) cumulative undersize curves of potato flour samples prepared under different processing conditions.

**Figure 4 foods-11-00507-f004:**
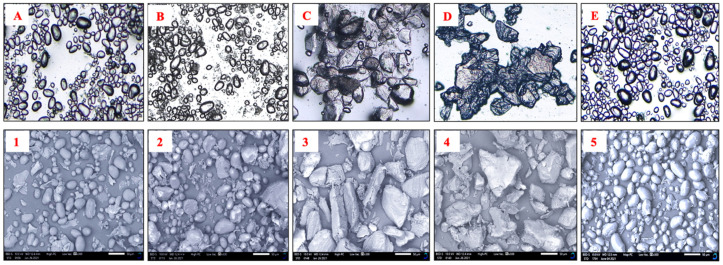
(**A**–**E**) Light microscopy (**1**–**5**) and scanning electron micrographs of potato flour prepared under different conditions. (**A**,**1**) Oven drying; (**B**,**2**) blanching at low temperature (60 °C); (**C**,**3**) blanching at high temperature (95°); (**D**,**4**) boiling; (**E**,**5**) freeze drying.

**Figure 5 foods-11-00507-f005:**
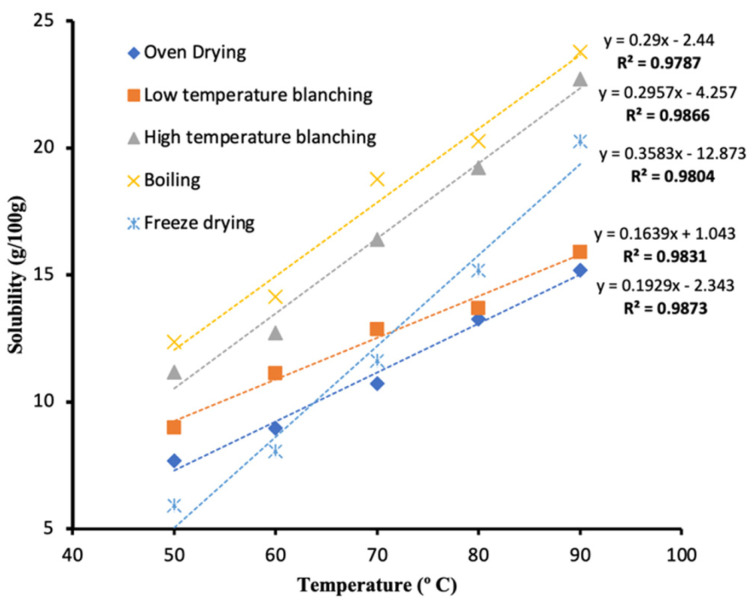
Solubility index of different potato flours at different temperatures.

**Figure 6 foods-11-00507-f006:**
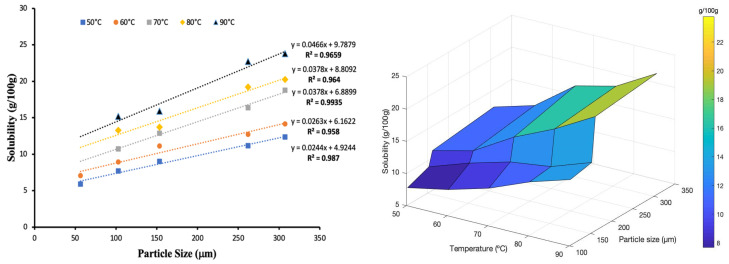
Correlation between particle size and solubility index.

**Table 1 foods-11-00507-t001:** Percentage yield of potato flour from three varieties as affected by pretreatments and drying methods.

Treatment	Shangi (%)	Unica (%)	Dutch Robjin (%)
Oven drying (OD)	18.48 ± 0.01 ^a^	17.31 ± 0.03 ^a^	22.50 ± 0.04 ^a^
Blanching (60°)_OD	17.25 ± 0.12 ^b^	16.31 ± 0.06 ^d^	20.83 ± 0.05 ^b^
Blanching (95°)_OD	15.77 ± 0.13 ^d^	16.01 ± 0.03 ^e^	20.42 ± 0.04 ^c^
Boiling_OD	16.50 ± 0.06 ^c^	16.83 ± 0.02 ^c^	19.44 ± 0.07 ^d^
Freeze-drying (FD)	18.70 ± 0.23 ^a^	17.64 ± 0.06 ^a^	22.99 ± 0.08 ^a^

Results are means of triplicate determinations ± standard error. Mean values with different letters in the same column indicate significant differences based on the LSD test of significance (*p* < 0.05, n = 3). OD, oven drying.

**Table 2 foods-11-00507-t002:** Color parameters of potato flour from different treatments and drying conditions.

Samples	L*	a*	b*	Chroma	Hue
Oven drying (OD)	90.86 ± 0.01 ^b^	0.31 ± 0.01 ^d^	15.19 ± 0.01 ^d^	15.19 ± 0.01 ^d^	1.55 ± 0.01 ^a^
Blanching (60°C)_OD	84.17 ± 0.01 ^c^	1.43 ± 0.01 ^c^	11.66 ± 0.01 ^e^	11.75 ± 0.01 ^e^	1.45 ± 0.01 ^c^
Blanching (95°C)_OD	82.36 ± 0.01 ^d^	1.71 ± 0.01 ^b^	20.72 ± 0.01 ^b^	20.79 ± 0.01 ^b^	1.49 ± 0.01 ^b^
Boiling_OD	80.38 ± 0.01 ^e^	4.07 ± 0.01 ^a^	25.90 ± 0.01 ^a^	26.22 ± 0.01 ^a^	1.42 ± 0.01 ^d^
Freeze-drying (FD)	92.86 ± 0.01 ^a^	−0.65 ± 0.01 ^e^	16.34 ± 0.01 ^c^	16.35 ± 0.01 ^c^	−1.53 ± 0.01 ^e^

Results are the means of triplicate determinations ± standard error. Mean values with different letters in the same column indicate significant differences based on the LSD test of significance (*p* < 0.05, n = 3). OD-Oven Drying.

**Table 3 foods-11-00507-t003:** Bulk density, tapped density, and true density of potato flour prepared under different pretreatment and drying conditions.

Samples	Bulk Density (g/mL)	Tapped Density (g/mL)	True Density (g/mL)
Oven drying (OD)	0.70 ± 0.01 ^c^	0.97 ± 0.02 ^b^	1.65 ± 0.01 ^a^
Blanching (60°C)_OD	0.77 ± 0.02 ^b^	1.01 ± 0.01 ^a^	1.64 ± 0.01 ^a^
Blanching (95°C)_OD	0.87 ± 0.02 ^a^	1.02 ± 0.01 ^a^	1.65 ± 0.00 ^a^
Boiling_OD	0.91 ± 0.02 ^a^	1.03 ± 0.01 ^a^	1.65 ± 0.01 ^a^
Freeze drying (FD)	0.49 ± 0.01 ^d^	0.70 ± 0.01 ^c^	1.46 ± 0.02 ^b^

Results are the means of triplicate determinations ± standard error. Mean values with different letters in the same column indicate significant differences based on the LSD test of significance (*p* < 0.05, n = 3). OD, oven drying.

**Table 4 foods-11-00507-t004:** Compressibility Index (CI) and angle of repose of potato flour.

Samples	CI (%)	Angle of Repose (°)	Flow Description	Type of Flour
Oven drying (OD)	27.73 ± 0.92 ^b^	30.57 ± 0.29 ^b^	Poor	Cohesive
Blanching (60 °C)_OD	21.79 ± 0.52 ^c^	28.94 ± 0.12 ^c^	Fair to good	Noncohesive
Blanching (95 °C)_OD	16.31 ± 0.61 ^d^	28.89 ± 0.17 ^c^	Good to excellent	Free-flowing (noncohesive)
Boiling_OD	13.48 ± 1.78 ^d^	28.01 ± 0.28 ^d^	Good to excellent	Free-flowing (noncohesive)
Freeze-drying (FD)	31.56 ± 1.07 ^a^	32.40 ± 0.12 ^a^	Very poor	Cohesive

Results are the means of triplicate determinations ± standard error. Mean values with different letters in the same column indicate significant differences based on the LSD test of significance (*p* < 0.05, n = 3). OD, oven drying. Flow behavior of potato flour was described according to a method developed by Carr [60].

**Table 5 foods-11-00507-t005:** Particle size parameters of potato flour samples.

Samples	*D_10_* (μm)	*D_50_* (μm)	*D_90_* (μm)	*Mean* (μm)
Oven drying (OD)	28.36 ± 0.33 ^c^	97.01 ± 0.09 ^d^	494.08 ± 1.36 ^d^	102.74 ± 4.23 ^d^
Blanching (60°)_OD	27.050.41 ^c^	233.49 ± 3.58 ^c^	562.93 ± 3.25 ^c^	153.31 ± 3.18 ^c^
Blanching (95°)_OD	95.55 ± 0.41 ^b^	310.41 ± 5.79 ^b^	668.76 ± 6.80 ^b^	261.87 ± 4.17 ^b^
Boiling_OD	105.14 ± 4.48 ^a^	344.09 ± 3.05 ^a^	708.07 ± 3.28 ^a^	307.53 ± 3.71 ^a^
Freeze-drying (FD)	17.59 ± 0.13 ^d^	52.87 ± 0.90 ^e^	182.77 ± 6.54 ^e^	56.51 ± 1.80 ^e^

Results are the means of triplicate determinations ± standard error. Mean values with different letters in the same column indicate significant differences based on the LSD test of significance (*p* < 0.05, n = 3). OD, oven drying. *D_10_*: 10% of granule volume diameter consists of smaller granules, *D_50_* granule: 50% of granule volume diameter consists of smaller granules, *D_90_*: 90% of granule volume diameter consists of smaller granules.

## Data Availability

Not applicable.

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
