# Peer review of "Effect of Pretreatments and Drying Methods on Physical and Microstructural Properties of Potato Flour"

_foods, 2022, doi:10.3390/foods11040507_

Round 1

Reviewer 1 Report

In this manuscript, “Effect of Pre-Treatments and Drying Methods on Physical and Microstructural Properties of Potato Flour”, the authors investigated the effects of pre-treatment and drying methods on the physical properties of three potato varieties. This manuscript was well-organized and suggested to be published if the following is considered.

The authors should review the current studies on the potential effects of pre-treatments and propose the potential challenge that needs to be addressed. The studies of dying methods (such as oven and frozen drying) will be also reviewed. It is better to highlight the current status, unknowns, and the contributions of this study to the scientific community.

L414-415: Please explain this strong positive correlation.

L425: Figure 6 is missing.

Author Response

Q1: Explain the strong correlation between particle size and solubility!

Answer:

 Our results indicate that the solubility of potato flour is correlated to particle size. This means that when the size of particles increases, the solubility increases in a similar way. The bigger size of the particles, the easier for them to dissolve. (Page 14, Line 442)

Observation:  Figure 6 missing

Action: Issue fixed in the manuscript. The sentence (429) was just a repetition of what was already said earlier, therefore, the sentence was deleted.

Reviewer 2 Report

The manuscript entitled "Effect of Pre-Treatments and Drying Methods on Physical and 2 Microstructural Properties of Potato Flour" has been evaluated. The results presented are considered to be very interesting and the quality of the presentation is very good. The results are relevant and the potato flour study was extensively analyzed. I consider that the analysis of the results are appropriate. I recommend that the results be a little more disputed in terms of similar results in other investigations.
In general terms, you should not make major changes to the manuscript, you just have to take into account some recommendations made in the document that will be attached to the review.

Author Response

Q1: How thick are the slices? were they made with an instrument or not?

Answer:

Potato tubers were washed with tap water, manually peeled, thinly sliced into 2-mm thickness. A sharp stainless-steel knife was used to slice the potato tubers

Q2: The wording is not clear. The same sample was subjected to two different temperatures and different times or does it refer to different samples?

Answer:

Different samples were subjected to different blanching treatments. A portion of potato flour was blanched at 60°C for 30 min and another one at 95° C for 1 min. (Page 3, Line 106)

Q3: What was the final moisture content in all cases?

Answer:

Moisture and other related proximate composition were not part of this manuscript. The moisture content ranged from 8.24% to 10.15% as shown in the table below.

Results for moisture content

Parameters

Oven Dry (OD)

Blanching (60°C_30Min)

Blanching

(95°C_1Min)

Boiling_OD

Freeze Drying (FD)

Moisture (%)

10.15±0.06b

10.33±0.02a

9.85±0,03c

8.24±0.02d

10.44±0.03a

Q4: Is it possible to put some potato flour pictures?

Answer:

Yes (see page 8)

Reviewer 3 Report

Comments and suggestions for authors

-English language and style require minor spell check required.

-In this study three varieties of potato have been compared, why is the data of the parameters analysed for the three varieties of potato not shown?

-More current references should be incorporated and added in the introduction section, especially when dealing with data on consumption and processing of potatoes.

Author Response

Q1: In this study, three varieties of potato have been compared, why is the data of the parameters analyzed for the three varieties of potato not shown?

Answer:

Similar trends were observed for the different varieties for different parameters studied. We decided to use Shangi as a reference.

Reviewer 4 Report

In this manuscript, the authors evaluated the effects of pre-treatments (blanching (60°C and 95°) and boiling) and drying methods (freeze-drying and oven drying) on the quality characteristics of potato flour. Following are my queries and suggestion:

  • What is the purpose of selecting oven drying and freeze drying. The operating charge for both the drying methods vary in many folds, and it eventually affects the quality of the product.
  • Potato flour study is an old study, and we can see plenty of published work in the past three decades. The novelty of this work is still missing?
  • Author can provide error bars for Figure 1.
  • Why only results for Shangi variety was used for the results and discussion?
  • What is the purpose of pretreatment and why it was conducted only for the oven drying and not freeze-drying?
  • I presume, both the drying techniques (oven drying and freeze-drying) will provide the dried potatoes in distorted structure. And it was powdered only by blender. Does the blender only determine the particle size?
  • Why moisture content was conducted immediately after drying. Because freeze-drying provide a very low moisture content than conventional oven drying. And apparently, the other physical properties are determined by the moisture content.

Author Response

Q1: What is the purpose of selecting oven drying and freeze-drying? The operating charge for both the drying methods varies in many folds, and it eventually affects the quality of the product.

Answer:

Oven and freeze-drying were chosen based on the preliminary work that was conducted earlier. Freeze drying and oven drying are known to produce products with acceptable physical properties such as color, shape, and size. Nutritionally, freeze-drying retained most of the nutritional values better than other drying methods. Yes, the cost of using freeze-drying is high but the product quality is excellent.

Q2. What is the purpose of pretreatment and why it was conducted only for the oven drying and not freeze-drying?

Answer:

Initially (preliminary work), blanching was conducted for both oven and freeze-drying. After some analyses were done, similar results were observed for the two treatments. Scanning electron microscopy images, particle sizes were not significantly different. Therefore, we decided to pick one for oven drying.

Q3. Why only results for Shangi variety was used for the results and discussion?

 Similar trends were observed for the different varieties for different parameters studied. We decided to use Shangi as a reference. (see Line 253)

Q4.  The author can provide error bars for Figure 1. (see page 7)

Q5: Novelty of the study

The novelty of this research is focused on understanding how the processing affects the quality of potato flour. Many studies have been conducted on potato flour but there is still no clear information on how the drying methods and pretreatments affect the microstructure and particle size of flour. The information on the correlation between particle size and physical and functional properties of potato flour is missing. These are important information as they predict the use of potato flour in different products development.

Round 2

Reviewer 4 Report

The author responded only to selected queries. And those provided responses were also not satisfactory. 

Author Response

Q1. I presume, both the drying techniques (oven drying and freeze-drying) will provide the dried potatoes in distorted structure. And it was powdered only by blender. Does the blender only determine the particle size?

Answer:

After grinding using a blender, potato flour was sieved through a 500 Micrometer standard sieve then stored for further analysis (See lines 109 and 117). The particle size was dependent on the processing conditions but also the intensity of crushing and shearing forces during blender milling. During crushing, potato granules can break apart (fractionate), becoming particulate material or stocks of different particle size distribution. Our results on SEM showed clearly these differences.